# Shift in Metabolite Profiling and Mineral Composition of Edible Halophytes Cultivated Hydroponically Under Increasing Salinity

**DOI:** 10.3390/metabo15110724

**Published:** 2025-11-05

**Authors:** Giedrė Samuolienė, Audrius Pukalskas, Akvilė Viršilė

**Affiliations:** Lithuanian Research Centre for Agriculture and Forestry, Akademija, 58344 Kėdainiai, Lithuania; audrius.pukalskas@lammc.lt (A.P.); akvile.virsile@lammc.lt (A.V.)

**Keywords:** carotenoids, controlled environment agriculture, edible halophytes, hydroponics, metabolite profiling, nutrient, organic acids, salinity, sugars, total proteins

## Abstract

Background: A significant concern today is the dependence on low-quality water sources, such as saline water, in hydroponic systems, especially due to the scarcity of freshwater. Halophytes and salt-tolerant species have emerged as viable candidates for cultivation in saline hydroponics. However, their agronomic performance and physiological responses within hydroponic systems require further investigation. Objectives: This research aims to explore the potential of edible halophytes grown in saline nutrient solutions within hydroponic systems within salt-tolerant ranges, focusing on their metabolic profiles and mineral accumulation. Methods: *Plantago coronopus* (L.), *Portulaca oleracea* (L.), and *Salsola komarovii* (Iljin) were grown in walk-in controlled environment chambers in deep water culture hydroponic systems, at 0, 50, 100, 150, and 200 mM·L^−1^ NaCl salinity; 16h, 250 µmol m^−2^ s^−1^, and wide LED spectrum lighting was maintained. Results: A significant decrease in organic acids, and fresh and dry weight under high saltinity was observed in *Plantago coronopus* and *Portulaca oleracea*, but not in *Salsola komarovii*. An increase in hexoses, particularly glucose, violaxanthin and β-carotene, P⁺ and Zn^2^⁺, along with a decrease in lutein, K⁺ and Ca^2^⁺ levels across salinity levels from 0 to 200 mM NaCl was observed in all treated halophytes. Increased salinity did not significantly affect total protein accumulation. Conclusions: These findings reveal that different shifts in osmolytes, mineral elements, and biomass accumulation in tested halophytes indicate species-dependent osmotic adjustment to increased salinity and may be attributed to the morphological differences among halophytic grasses, dicot halophytes, and those with succulent leaves or stems. The PCA score scatterplot results excluded the response of *Plantago coronopus* from other tested halophytes; also, it demonstrated that *Portulaca oleracea* was more sensitive to the hydroponic solution salinity compared to *Salsola komarovii* and *Plantago coronopus*.

## 1. Introduction

As global food demand rises and environmental challenges become more pressing, sustainable hydroponic agriculture is emerging as an effective solution to minimize water usage [1], improve resource efficiency [2], and detach food production from the limitations of arable land [3,4]. In this framework, Controlled Environment Agriculture (CEA) offers notable benefits through the precise management of climate and nutrient factors, enabling continuous cultivation throughout the year while reducing environmental footprints [5]. Recirculating hydroponic systems, a crucial component of CEA, facilitate the recycling of water and nutrients, thereby reducing waste and maximizing resource utilization [6]. However, maintaining high productivity levels under these resource-conserving regimes remains a critical challenge. The heightened interest in CEA and hydroponic systems reflects the urgent need for sustainable food production methods, particularly in areas facing water scarcity [7]. Despite their advantages in conserving water and nutrients, ensuring consistent crop productivity and quality within these systems remains a critical factor that needs to be addressed.

A significant concern today is the reliance on low-quality water sources, such as saline water in hydroponic systems, especially given the scarcity of freshwater [8]. Salinity stress negatively impacts plant growth and development [9], ionic balance [8,10], and metabolic processes [11]. This challenge has prompted interest in halophytes and salt-tolerant crops, which are naturally suited to saline conditions and exhibit unique physiological and biochemical traits [12]. In response, attention has turned to halophytes and salt-tolerant crops, which naturally thrive in saline environments and possess unique physiological and biochemical adaptations. Halophytes are salt-tolerant plants that can complete their life cycle in a salt concentration of at least 200 mM NaCl, and some extreme halophytes survive even under higher than 1000 mM NaCl concentrations [13]. 

*Plantago* L. is a genus in the Plantaginaceae family, with about 300 species. The species is characterized by its numerous epilictic leaves forming a dense flat rosette and spikes bearing small greenish flowers [14,15]. Due to its rich phytochemical composition, including polysaccharides, flavonoids, phenolic acids, glycosides, lignans, triterpenes, vitamins, and mineral elements [16], *Plantago coronopus* is commonly used as a medicinal plant or as a vegetable and has potential as a cash crop for animal feed. Extracts are rich in total polyphenols (˃20 mg·g^−1^), flavonoids (0.10–146 mg RE g^−1^), antioxidant activity (0.32−˃10 mg mL^−1^), amino acids, such as arginine (10.2 mg·g^−1^ DW), leucine (5.6 mg·g^−1^ DW), threonine (5.2–45.1 mg·g^−1^ DW), mineral elements, such as Na, Ca, K, Mg, Fe, Mn, Zn (50.0, 14.0, 8.0, 6.3, 0.4, 0.01, 0.05 mg·g^−1^ DW) [17].

*Portulaca oleracea* L. (common purslane) is an annual succulent herb of the Portulacaceae family. The plant exhibits a prostrate, fleshy habit with smooth reddish stems, obovate leaves, and small yellow flowers borne singly or in clusters at stem apices. Its high adaptability to arid and saline environments is attributed to its C_4_; photosynthetic pathway and efficient water-use physiology [18]. *P. oleracea* is rich in both primary and secondary metabolites of nutritional and pharmacological importance. The leaves contain proteins (18–25%), carbohydrates (35–45%), and dietary fiber (10–12%), alongside minerals such as potassium (3.5–5.0 g·100 g^−1^ DW), calcium (1.5–2.2 g·100 g^−1^), magnesium, and iron. It is notably one of the richest terrestrial sources of α-linolenic acid (up to 400 mg·g^−1^ of total fatty acids). Major secondary metabolites include phenolic acids (caffeic, ferulic, and p-coumaric acids), flavonoids (kaempferol, quercetin, and apigenin derivatives), betalains, alkaloids (oleraceins), and vitamins C and E [19]. These compounds confer potent antioxidant, anti-inflammatory, and neuroprotective properties, supporting its traditional use as both a medicinal and highly nutritious edible plant.

*Salsola komarovii* Iljin is an annual halophytic herb belonging to the Amaranthaceae family. It is a succulent plant with cylindrical, fleshy leaves and small, inconspicuous flowers in axillary clusters [20]. *S. komarovii* exhibits a diverse phytochemical profile dominated by phenolics (10–25 mg·g^−1^ DW), flavonoids (kaempferol, quercetin, and isorhamnetin glycosides reach 4–8 mg·g^−1^ DW), and minerals such as sodium (5–7 g·100 g^−1^ DW), potassium (2–3 g·100 g^−1^), calcium, magnesium, and iron. Betaine (up to 25 mg·g^−1^ DW) and pinitol act as major osmolytes. Vitamins C and E, carotenoids, and small amounts of saponins and sterols further enhance its bioactive potential [21]. Research indicates that the content of phytochemicals is highly dependent on the developmental stage of the plant, organ (leaves, stem, or roots), and environmental conditions.

Studies revealed that several *Plantago* species display a range of salt tolerance levels, positioning them as outstanding models for insightful comparative studies on salinity stress responses [22]. *Plantago coronopus* is notable not for its high salt tolerance, but for its unique growth and reproductive traits [23]. *Salsola* species shows significant promise as a model plant for exploring cross-tolerance to salt and drought stress, offering important insights that could enhance stress resistance in various other species [24,25]. Additionally, its phytochemical compounds, such as alkaloids and flavonoids, may have valuable pharmaceutical and nutritional applications [26]. *Portulaca oleracea* L. is also classified as an eu-halophyte, a highly salt-tolerant plant [27], valuable for its antioxidant compounds, vitamins, and minerals [18,28]. However, the integration of these plants into hydroponic systems remains limited, primarily due to the absence of specific cultivation protocols and optimization methods. One compelling approach is to use saline or marginal-quality water in hydroponic setups [29,30]. Yet, salinity poses significant stress on plant growth, ion balance, and metabolism [31].

Moreover, salinity-induced stress triggers the synthesis of secondary metabolites, including phenolics, terpenoids, and nitrogen (such as alkaloids)- and sulfur-containing (e.g., glucosinolates) compounds, which play essential roles in plant defense and contribute to human health benefits [32,33]. However, the relationship between salinity levels, secondary metabolite biosynthesis, mineral element availability, and plant growth trade-offs under hydroponic conditions remains poorly defined. Additionally, most nutrient solution formulations have been developed for glycophytic species, leaving a gap in tailored nutrient regimes for salt-resilient crops. Furthermore, most nutrient solution formulations are designed for glycophytic crops, leading to a lack of tailored mineral nutrition strategies for halophytes or salt-tolerant horticultural species. This knowledge gap hinders optimization of both biomass production and metabolite accumulation. Despite increasing interest in sustainable hydroponics and CEA, there are not many integrative studies linking plant growth, salinity adaptation, and secondary metabolism. There is a critical need to develop crop- and context-specific models that account for physiological and metabolic responses under constrained water and nutrient scenarios. Thus, halophytes and salt-tolerant species have emerged as viable candidates for saline hydroponic cultivation. Nonetheless, their agronomic performance and physiological responses within recirculating hydroponic systems still require further investigation.

Considering the identified gaps, this research seeks to compare the potential of edible halophytes, such as *Plantago coronopus*, *Portulaca oleracea*, and *Salsola komarovii* grown in saline nutrient solutions in hydroponic systems within salt-tolerant ranges, focusing on their metabolic profile and mineral accumulation.

## 2. Materials and Methods

### 2.1. Plant Material and Experimental Setting

*Plantago coronopus* (L.), *Portulaca oleracea* (L.)*,* and *Salsola komarovii* (Iljin) seeds were purchased from CN seeds (Pymoor, Ely, Cambridgeshire, UK), and germinated in water-soaked rockwool cubes (20 × 20 mm). On the 10th day after germination, seedlings were transferred into deep water culture (DWC) hydroponic systems. DWC tanks of 40 L volume were used for the experiment. Each tank represents an experimental replication, containing 12 mesh pots, *Plantago coronopus* and *Portulaca oleracea*–five plants, and *Salsola komarovii* one plant in each. Plant samples were analyzed after 3 weeks from germination for *Plantago coronopus* and *Portulaca oleracea*, and after 4 weeks for *Salsola komarovii*. Nutrient solution concentrate was provided by Plagron (Hydro A&B, Ommelpad, The Netherlands). The salinity concentrations of hydroponic solutions were maintained at 0, 50, 100, 150, and 200 mM·L^−1^ NaCl. All experiments were performed in walk-in chambers, maintaining +21/17 °C temperature and ~65% relative air humidity. Wide-spectrum lighting: equal spectral composition of deep red 61%, blue 20%, white 15%, far-red 4% (TUAS GTR 2V 0021096109 C1 DL ST, Tungsram, Hungary) was used, and photosynthetic photon flux density (PPFD) of 250 µmol m^−2^ s^−1^ was maintained [28,34]. PPFD was measured and regulated at the plant level using a photometer–radiometer (RF-100, Sonopan, Poland).

### 2.2. Measurements

Plant material was freeze-dried, ground using an ultra-centrifugal mill ZM 300 (Retsch GmbH, Haan, Germany) with ring sieve pore size 0.5 mm, and stored till analysis.

#### 2.2.1. HPLC Analysis of Sugars and Organic Acids

Sample preparation for sugars and organic acids analysis: 0.1 g of freeze-dried plant material was mixed with 4 mL of warm (40–50 °C) deionized water and then rotated for 2 h on the LABINCO LD79 digital test tube rotator (Labinco BV, Breda, The Netherlands). The tubes were centrifuged at 4500 rpm in a Z366K centrifuge (HERMLE Labortechnik GmbH, Wehingen, Germany) for 15 min. Then, 1 mL of supernatant was filtered through a 0.22 μm nylon syringe filter and transferred to a chromatography vial for organic acid analysis. A total of 0.9 mL of supernatant was transferred into a 2 mL microtube and mixed with 0.9 mL of 0.01% ammonium acetate solution in acetonitrile and stored in a refrigerator at 4 °C for 30 min. for separation of organic acids. Samples were centrifuged at 14,000 rpm for 15 min. in a MiniSpin centrifuge (Eppendorf AG, Hamburg, Germany), filtered through 0.22 μm nylon syringe filters to chromatography vials for analysis of sugars.

HPLC analyses of organic acids were performed on NEXERA HPLC (Shimadzu, Japan) with LCMS 2020 mass detector using YMC-Triart C18 column (150 × 3 mm, 3 µm) (YMC Europe GmbH, Dinslaken, Germany) as described by Flores et al. [35].

HPLC analyses of sugars performed on NEXERA HPLC (Shimadzu, Japan) with ELSD-LTII detector, using Shodex SUGAR SP0810 (300 × 8 mm, 7 µm) column (Shova Denko Europe GmbH, Germany), the separation was performed at +85 °C, ultra-pure water (resistivity 18.2 MΩ∙cm; TOC—1 ppb) was used as mobile phase [34].

#### 2.2.2. HPLC Analysis of Carotenoids and Chlorophylls

Sample preparation for carotenoids and chlorophylls analysis: about 0.05 g of freeze-dried ground plant material was mixed with 3 mL of 80% acetone. The extraction was carried out for 24 h at +4 °C temperature. Extracts were centrifuged at 10,000 rpm for 10 min and filtered through a 0.22 μm PTFE (hydrophilic) syringe filter (VWR International, Radnor, PA, USA).

The HPLC analysis of carotenoids was conducted using an SDP-M10A DAD detector. This method follows the protocol outlined by Sander et al. [36], with modifications made as described by Samuolienė et al. [34] on YMC C30 carotenoid column (250 × 3 mm, 5 µm), using acetone: methanol (50:50) as mobile phase.

#### 2.2.3. Spectrophotometry-Based Protein Estimation

Total protein content was evaluated spectrophotometrically using a microplate reader (SpectroStarNano, BMG Labtech, Ortenberg, Germany) as described by Bradford [37]. Analysis was performed by adding Folin–Ciocalteu reagent and 10% sodium carbonate (Na_2_CO_3_) solution to the sample extract. After 20 min., the absorbance of the mixture was measured at 765 nm. The contents of TPC were quantified as gallic acid equivalents according to the calibration curve. Results were expressed by mg of TPC g^−1^ DW.

#### 2.2.4. Spectrometry-Based Elemental Analysis

The contents of macro (P, K, Ca, Mg)- and microelements (Fe, Mn, Na, Zn) were determined using the microwave-assisted digestion technique (Multiwave GO, Anton Paar, Graz, Austria), combined with inductively coupled plasma optical emission spectrometry (Spectro Genesis, Kleve, Germany), as described by Araújo et al. [38] and Barbosa et al. [39]. Complete mineralization of 0.3 g dry plant material was achieved with 8 mL 65% HNO_3_. The digestion program was as follows: (1) 170 °C reached within 3 min, digested for 10 min; (2) 180 °C reached within 10 min, digested for 10 min. Mineralized samples were diluted to 50 mL with deionized water. The calibration curves for all the studied elements were in the range of 0.01–400 mg L^−1^.

#### 2.2.5. Biometric Measurements

Fresh weight (FW g), dry weight (DW g), and dry matter (DW/FW, expressed in %) were determined.

### 2.3. Statistical Analysis

Data were processed using XLStat (2025) software, using one-way ANOVA, Tukey’s HSD at the confidence level *p* < 0.05, and Pearson’s correlation analysis. For result modeling, principal component analysis (PCA) was performed.

## 3. Results

### 3.1. Metabolites

Organic acid dynamics under different levels of hydroponic salinity were species-specific (see Table 1). An increase in the salinity of the hydroponic solution had no significant effect on oxalic acid accumulation. However, a significant decrease in the content of malic acid (1.2 to 1.9 times), fumaric acid, succinic acid (more than 2 times), and ascorbic acid (more than 7 times) was observed in *Plantago coronopus*. In *Portulaca oleracea*, raising the NaCl concentration from 100 to 200 mM·L^−1^ resulted in a significant decrease in oxalic, succinic, and citric acids by 30%, 21%, and 24%, respectively, along with about a 50% reduction in malic and fumaric acids. In contrast, a salinity level of 50 to 100 mM·L^−1^ led to a significant increase in ascorbic acid in *Portulaca oleracea*. Contrary to the mentioned plants, increasing the NaCl concentration in *Salsola komarovii* resulted in significant increases in oxalic, citric, malic, and fumaric acids. However, a salinity level of 100 to 200 mM·L^−1^ resulted in a significant decrease in succinic acid, and ascorbic acid was not detected.

The increasing salinity level of the hydroponic solution led to sugar accumulation in *Plantago coronopus* and *Portulaca oleracea*, but had no significant effect on sugar accumulation in *Salsola komarovii* (Figure 1A). *Plantago coronopus* accumulated the highest amounts of sugars, particularly due to significant glucose accumulation (43.6–78.8 mg·g^−1^ DW) (Figure 1A), while *Salsola komarovii* was noted for the highest protein (78.4–84.5 mg·g^−1^ DW), lutein (350–470 μg·g^−1^ DW) and β-carotene (310–787 μg·g^−1^ DW) content (Figure 1C). As salinity levels of the hydroponic solution increased, there was a significant increase in glucose, up to 2 times, in *Plantago coronopus* and both fructose (up to 2.7 times) and glucose (up to 2.2 times) in *Portulaca oleracea*. Furthermore, NaCl concentrations of 150 mM and 200 mM led to the accumulation of raffinose in *Plantago coronopus*, while concentrations of 100 mM to 200 mM NaCl resulted in maltose accumulation in *Portulaca oleracea* (Figure 1A). Significantly the lowest total protein content was observed at salinity levels of 0 mM and 200 mM NaCl in *Plantago coronopus* and *Portulaca oleracea,* respectively (Figure 1C). 

Salinity levels of 50 mM to 150 mM NaCl significantly increased the accumulation of lutein and β-carotene in *Plantago coronopus* and *Portulaca oleracea*. In contrast, *Salsola komarovii* exhibited the highest lutein levels at 0 and 50 mM NaCl, 468.4 and 443.2 μg·g^−1^ DW, respectively. While its accumulation of violaxanthin (about 70 μg·g^−1^ DW) and β-carotene (530.8–786.2 μg·g^−1^ DW) was significantly higher at salinity levels of 150 mM and 200 mM NaCl (Figure 1B).

### 3.2. Biomass and Mineral Elements

The increasing salinity level from 0 mM to 200 mM of NaCl resulted in a significant decrease in FW and DW in both *Plantago coronopus* (up to 3.3 times FW, and 2.7 times DW) and *Portulaca oleracea* (up to 6.2 times FW, and 4.8 times DW) (Figure 2D). The salinity level from 50 mM to 150 mM led to increased FW (about 25–27 g) and DW (2.0–2.2 g) in *Salsola komarovii*. Dry matter accumulation was as follows: *Portulaca oleracea* ˃ *Plantago coronopus* ˃ *Salsola komarovii* (Figure 2E). 200 mM salinity level resulted in a significant increase in dry matter (about 20%) in *Portulaca oleracea* and *Plantago coronopus*. Compared to lower hydroponic solution salinity, 150–200 mM NaCl concentrations also led to an increase in dry matter in *Salsola komarovii* (approximately 16.2%).

A significant increase in K, Ca, Mg, and Na was observed only in the control, the salinity levels did not significantly affect the accumulation of tested mineral elements in *Plantago coronopus* (Table 2). In contrast, both *Portulaca oleracea* and *Salsola komarovii* exhibited significant increases in P content, by 2.5 and 2.0 times, respectively, compared to the control. Additionally, Zn levels increased significantly by 4.4 times in *Portulaca oleracea* and by 1.2 times in *Salsola komarovii* as salinity levels rose. Nonetheless, the increased salinity in the hydroponic solution led to a decrease in the levels of other macro and microelements in both *Portulaca oleracea* and *Salsola komarovii*.

### 3.3. Correlation and Principal Component Analysis

Significant positive correlations between salinity, dry matter, soluble sugars, and Zn, and significant negative correlations between salinity, fresh weight, dry weight, organic acids, total proteins, K, and Ca were found in *Plantago coronopus* and *Portulaca oleracea* (Figure 3A,B, Appendix A). While in *Salsola komarovii* significant positive correlations between salinity, citric acid, β-carotene, P, and Zn, and significant negative correlations between salinity, raffinose, succinic acid, lutein, K, Ca, Mg, Mn, and Na were detected (Figure 3C, Appendix A). Significant positive correlations in *Plantago coronopus* and *Portulaca oleracea* were found between K, Ca, organic acids and total proteins, and significant negative correlation between K, Ca and soluble sugars was observed (Figure 3A,B, Appendix A). Significant positive correlations were found between Na and organic acids in *Plantago coronopus* (Figure 3A, Appendix A), while negative correlations were observed between Na and sugars. In contrast, *Portulaca oleracea* exhibited a strong positive correlation between Na and hexoses (Figure 3B, Appendix A). In *Salsola komarovii*, there were significantly strong positive correlations among K, Ca, Na, raffinose, succinic acid, and lutein, along with significantly negative correlations between K, Ca, Na, citric acid, and violaxanthin (Figure 3C, Appendix A). Additionally, both *Plantago coronopus* and *Portulaca oleracea* showed significantly negative correlations between soluble sugars, organic acids, and total proteins (Figure 3A,B, Appendix A). However, in *Salsola komarovii*, the negative correlation was only noted between soluble sugars and oxalic acid (Figure 3C, Appendix A).

The PCA score scatterplot (Figure 4) shows the average coordinates of sugars, organic acids, total protein contents, mineral elements, and biomass accumulation in *Plantago coronopus, Portulaca oleracea*, and *Salsola komarovii* grown under different hydroponic solution salinity levels. F1 component explained 39.84% of the total variability and demonstrated the parallel role of organic acids, mineral elements (K, Ca, Mg, Fe), glucose, fresh and dry weight, and violaxanthin (Figure 4), which excluded the response of *Plantago coronopus* from other tested halophytes. The response of *Salsola komarovii* distinguished from others according to F2 (24.7%) by total protein, lutein, and β-carotene contents. The PCA score scatterplot results also demonstrate that *Portulaca oleracea* was more sensitive to the hydroponic solution salinity compared to *Salsola komarovii* and *Plantago coronopus*.

## 4. Discussion

### 4.1. Differential Metabolic Responses of Halophytes to Increased Salinity

Higher salinity causes significant ionic, osmotic, and metabolic imbalances, resulting in detrimental alterations to the physiological and molecular aspects of cellular components. In response to these conditions, plants regulate various metabolites that help protect them from the effects of salinity [40]. It is known that halophyte salt-tolerance arises from an integrated network of signaling systems that collectively reduce stress impacts [41] through regulation of ion exchange [42], enhanced activity of antioxidants [43], and increased accumulation of sugars [44] and organic acids [45]. However, it has been reported that in halophytic species such as *Puccinellia tenuiflora*, organic acids have been found to decrease under salt stress [46]. While Samuoliene et al. [35] found that increased salt concentration resulted in increased levels of citric, malic, and fumaric acids, decreased levels of oxalic acid, and no significant effect on succinic acid in *Mesembryanthemum crystallinum*. Our results indicate that the accumulation of organic acids can be species-dependent, as a significant decrease with increasing salinity was observed and confirmed by a significantly strong negative correlation in *Plantago coronopus* and *Portulaca oleracea*, but not in *Salsola komarovii* (Table 1, Figure 3). The variations in organic acid accumulation among different halophyte species subjected to salt stress may be attributed to the morphological differences in their specialized salt-secreting structures. These structures play a crucial role in excreting or accumulating excess salts from metabolically active tissues, such as succulent leaves or stems [13,47]. The accumulation of sugars has been linked to mechanisms of tolerance for salinity and drought. The increase in compatible solutes enhances cellular osmolarity, drawing water into the cell and helping preserve turgor for cell expansion [45]. It has been reported that sucrose, glucose, and fructose accumulate in response to salinity, with sucrose present in higher amounts in *Thellungiella halophila* [48], *Juncus acutus* and *J. maritimus* [44]. An increase in glucose was determined in *Mesembryanthemum crystallinum* under 100–200 mM hydroponic solution salinity [34]. We observed similar results, with an increase in hexoses, particularly significant glucose accumulation, as hydroponic solution salinity increased (Figure 1A), and a significantly strong positive correlation between salinity and soluble sugars was found in *Plantago coronopus* and *Portulaca oleracea*, but not in *Salsola komarovii* (Figure 3). This increase in glucose serves as an effective adaptation strategy to conserve energy during salt stress, thereby allowing halophytes to allocate energy toward active stress tolerance. Additionally, salinity-induced accumulation of individual soluble sugars such as raffinose and maltose was observed in *Plantago coronopus* and *Portulaca oleracea,* respectively. Meanwhile, in *Salsola komarovii*, raffinose was detected in a no-salt (0 mM NaCl) hydroponic solution (Figure 1A), indicating that the role of individual sugars in salt resistance among halophyte species might be different. Moreover, signal molecules such as sucrose, fructose, raffinose, and trehalose can activate the immune system of plants. Raffinose and galactinol also serve as non-enzymatic antioxidants, helping to degrade reactive oxygen species [49,50]. The accumulation of various soluble proteins has developed into a crucial strategy that significantly regulates the growth and development of plants experiencing salt stress. Furthermore, proteins play a crucial role in salt tolerance, more so than gene expression, as they are essential in defining the physiological responses of salt-tolerant phenotypes. [51]. Proteins regulate the uptake and transport of Na^+^ and K^+^ ions, activating enzymes that scavenge reactive oxygen species (ROS) and improve osmotic adjustment under normal conditions as well as during salt stress [52]. Little information is available on salt stress-responsive proteins. However, Dissanayake et al. [53] indicate that the quality and type of proteins could be more crucial for salt tolerance than the total soluble proteins. Our results indicate that salinity stress does not significantly affect total protein accumulation in treated halophytes (Figure 1A), whereas a significantly strong negative correlation between salinity and total proteins in *Plantago coronopus* and *Portulaca oleracea*, but not in *Salsola komarovii* was observed (Figure 3). Qun et al. [54] found increased accumulation of salt stress protein osmotin in *Mesembryanthemum crystallinum* under salt stress. Changes in the levels of carotenoids in halophytes due to salinity have been documented [55]. In agreement with Fitzner et al. [55], we observed an accumulation of violaxanthin and β-carotene, and a decrease in lutein within salt ranges of 0 (no salt)—200 mM NaCl (Figure 1B and Figure 3). For *Salsola komarovii*, we observed particularly low contents of violaxanthin and about 1.5 times higher lutein and β-carotene contents, compared to *Plantago coronopus* and *Portulaca oleracea* in a no-salt hydroponic solution.

### 4.2. Ion Regulation and Biomass Allocation Strategies in Halophytes Under Increased Salinity

It is presumed that halophytes may adapt to increased salinity through distinct regulation of ion transport, such as Na^+^, Cl^−^, and K^+^ ions, which could involve variations in their photosynthetic apparatus. Research indicates that as salinity increases, halophyte grasses typically reduce their above-ground biomass more significantly than their below-ground biomass, showing less reliance on the succulence strategy. In contrast, dicot halophytes exhibit a wide variation in stem and leaf succulence depending on the salinity levels in the root zone [56]. We observed a significant decrease in fresh and dry weight for *Plantago coronopus* and *Portulaca oleracea* with increased salinity. Meanwhile, salinity treatment had no significant effect on biomass changes in *Salsola komarovii* within salt-tolerant ranges (Figure 2D and Figure 3). Moreover, the different correlation patterns between fresh weight and osmolytes, such as soluble sugars, organic acids, and total proteins, differed in *Salsola komarovii*, compared to other treated halophytes (Figure 3), suggesting distinct species-specific osmotic adjustment associated with metabolite synthesis. Higher levels of sugars and a decrease in total proteins can be a good strategy for halophytes to increase salt tolerance. It is known that salinity increases Na^+^ accumulation in plants while decreasing K^+^ levels [57]. Meanwhile, halophytes use specialized mechanisms to maintain a higher K^+^/Na^+^ ratio for salt tolerance. These mechanisms include compartmentalizing extra ions in the vacuole and producing osmolytes to reduce cytosolic toxicity [58]. The plasma membrane of the cell is damaged by saline ions, causing mineral ions like P^+^, Mg^2+^, K^+^, and Ca^2+^ to exit the cell while external saline ions such as Na^+^ and Cl^−^ enter. This disrupts the cell membrane’s physiological function, affects the transport of Na^+^, K^+^, and Ca^2+^, alters ion content in the leaves, and results in irreversible damage to leaf functions [59]. Meanwhile, *Plantago coronopus, Portulaca oleracea*, and *Salsola komarovii* demonstrated the gradual increase of P+ and Zn^2+^ as salinity increased (Table 2). In agreement with Animasaun et al. [60], increasing the hydroponic solution salinity resulted in a gradual Na^+^ decrease, except in *Portulaca oleracea* Na^+^ significantly increased in 200 mM NaCl (Table 2). Indicating that halophytes successfully manage higher salinity by eliminating excess Na^+^ till their salt-tolerant ranges. On the other hand, all tested halophytes exhibited a gradual reduction in K^+^ and Ca^2+^ content as NaCl concentration increased (Table 2, Figure 3). Significant negative correlation between K^+^, Ca^2+^ and soluble sugars, such as glucose and maltose in *Portulaca oleracea* and glucose and raffinose in *Plantago coronopus* (Figure 3) supports the statement that glucose plays a crucial role in the osmotic adjustment of various plant species. Meanwhile, maltose enhances the protective effects on cell membranes, allowing normal cell function to be maintained during osmotic stress [61]. Hence, halophytes possess a complicated mechanism for salt tolerance, allowing them to control Na^+^ accumulation and maintain K^+^ and Ca^2+^ levels even in fluctuating salinity environments [47]. This capacity has its limits, as an excessively high concentration of salt stress can directly impair the selective absorption ability and lead to irreversible harm to plants.

## 5. Conclusions

A significant decrease in organic acids, amd fresh and dry weight under higher hydroponic solution salinity was observed in *Plantago coronopus* and *Portulaca oleracea*, but not in *Salsola komarovii*. An increase in hexoses, particularly glucose, violaxanthin and β-carotene, P^+^ and Zn^2+^, along with a decrease in lutein, K^+^ and Ca^2+^ levels across salinity levels from 0 to 200 mM NaCl was observed in all treated halophytes. Increased salinity did not significantly affect total protein accumulation. These findings reveal that the role of individual metabolites, particularly organic acids and proteins, in salt resistance among halophytes differs and may be attributed to the morphological differences among halophytic grasses, dicot halophytes, and those with succulent leaves or stems. Different shifts in osmolytes, mineral elements, and biomass accumulation in *Pantago coronopus* and *Portulaca oleracea* compared to *Salsola komarovii* indicate species-dependent osmotic adjustment to increased salinity. More research is required to understand the mechanistic basis of species-specific osmotic adjustment in halophytes by integrating metabolomic and transcriptomic approaches. Comparative analyses among different halophyte sensitivity to salinity would provide insights into the differential regulation of primary and secondary metabolites, and mineral ions under salt stress. Moreover, investigating the relationship between tissue morphology, ion compartmentalization, and photosynthetic performance could further clarify the adaptive strategies conferring salt tolerance in diverse halophytic species.

## Figures and Tables

**Figure 1 metabolites-15-00724-f001:**
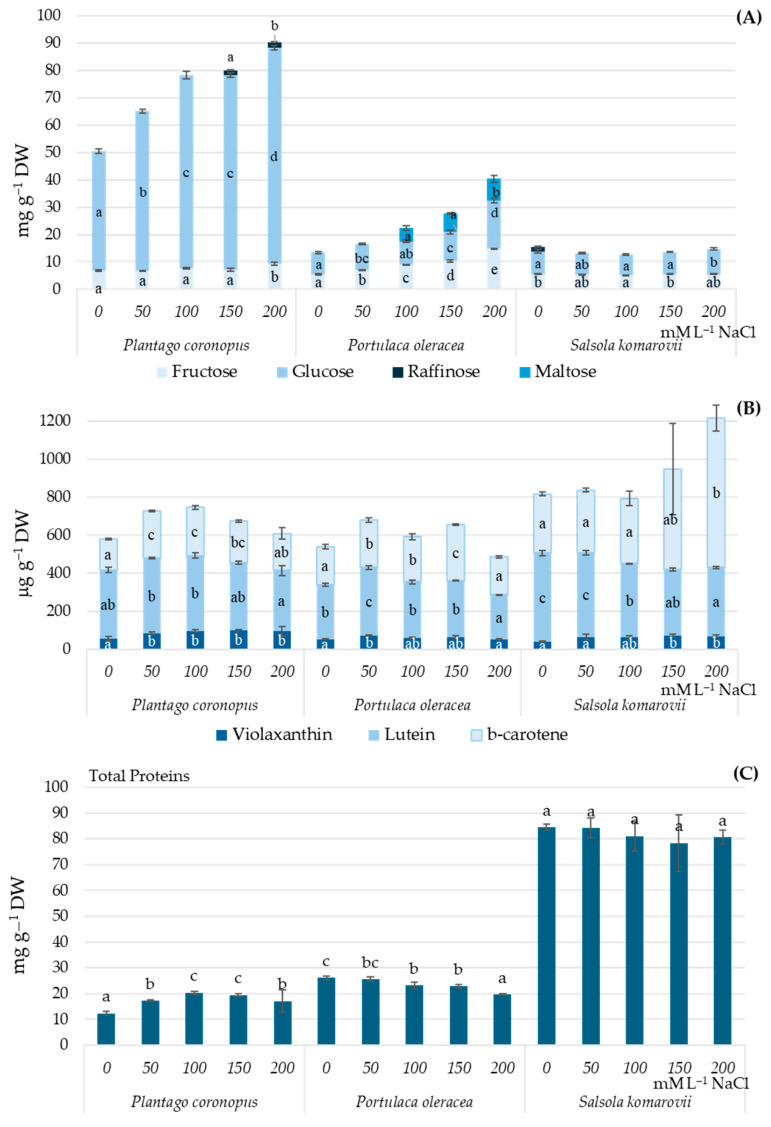
Soluble sugars (**A**), carotenoids (**B**), and total protein (**C**) contents in *Plantago coronopus, Portulaca oleracea,* and *Salsola komarovii* shoots depending on the hydroponic solution salinity (0–200 mM·L^−1^ NaCl). Different letters indicate statistically significant differences between means of salinity treatments according to one-way ANOVA Tukey’s test when *p* < 0.05.

**Figure 2 metabolites-15-00724-f002:**
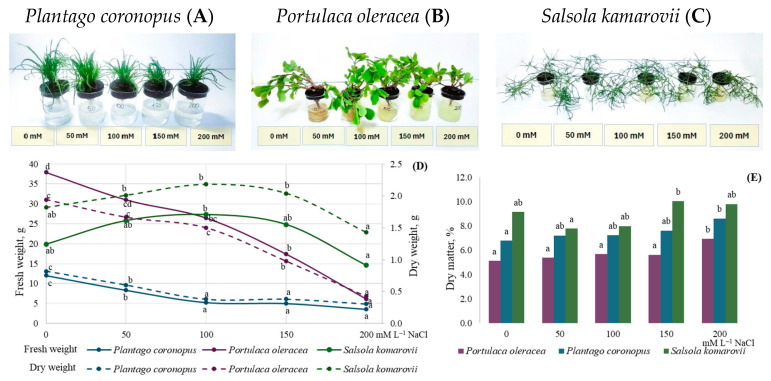
Fresh and dry weight (**D**), dry matter (**E**) accumulation in *Plantago coronopus* (**A**), *Portulaca oleracea* (**B**), and *Salsola komarovii* (**C**) shoots, depending on the hydroponic solution salinity (0–200 mM·L^−1^ NaCl). Different letters indicate statistically significant differences between means of salinity treatments according to one-way ANOVA Tukey’s test when *p* < 0.05.

**Figure 3 metabolites-15-00724-f003:**
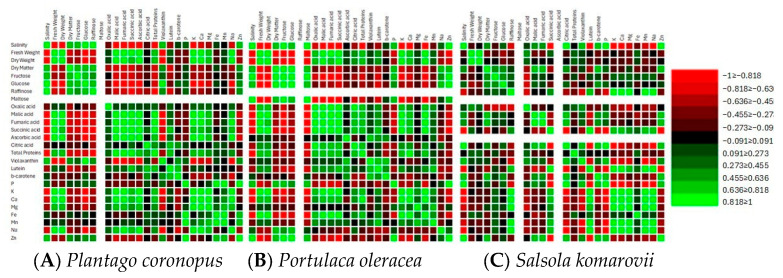
Pearson’s correlation analysis between physiological parameters under different salinity in *Plantago coronopus* (**A**), *Portulaca oleracea* (**B**), and *Salsola komarovii* (**C**). Each cell color is different from 0 with a significance level α = 0.05.

**Figure 4 metabolites-15-00724-f004:**
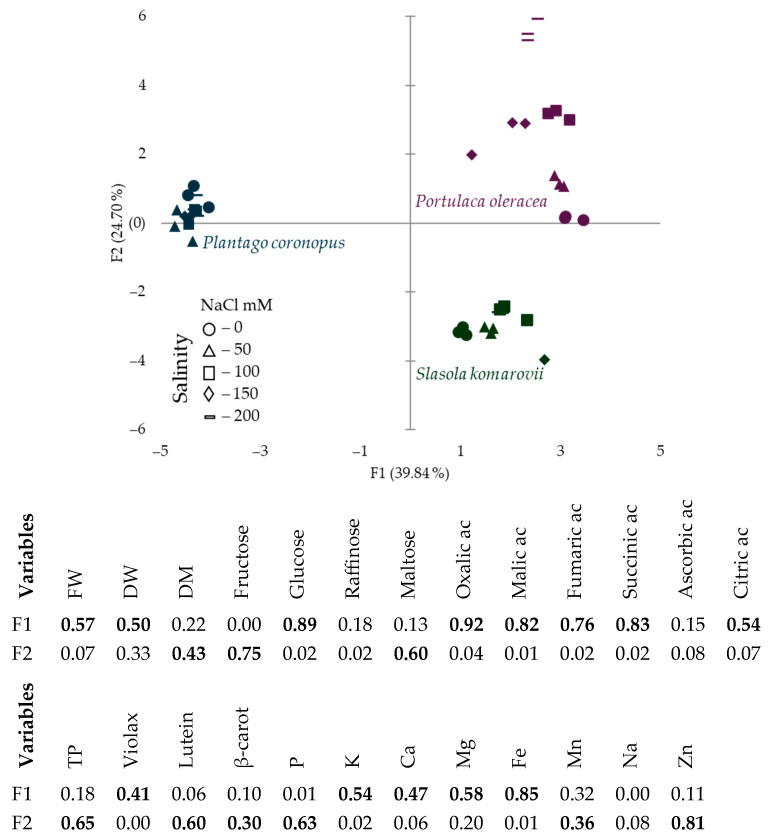
The scatterplot of principal component analysis (PCA) and distribution of squared cosines of the variables illustrates the relationships between phytochemical composition, mineral elements, and growth response depending on hydroponic solution salinity in *Plantago coronopus, Portulaca oleracea,* and *Salsola komarovii*. Values in bold correspond for each variable to the factor for which the squared cosine is the largest. FW—fresh weight, DW—dry weight, DM—dry matter, TP—total proteins, Violax—Violaxanthin, β-caroten—β-carotene.

**Table 1 metabolites-15-00724-t001:** The effect of hydroponic solution salinity on the accumulation of organic acids in *Plantago coronopus, Portulaca oleracea* and *Salsola komarovii* leaves (mg·g^−1^ DW).

NaCl mM·L^−1^	Oxalic Acid	Malic Acid	Fumaric Acid	Succinic Acid	Ascorbic Acid	Citric Acid
*Plantago coronopus*						
0	0.029a	44.5d	52.6d	1.22c	0.361d	3.82a
50	0.025a	38.3c	41.4c	1.17bc	0.223c	3.95a
100	0.029a	30.0b	29.8b	1.13b	0.054b	4.42b
150	0.025a	28.8b	28.6b	0.96a	0.000a	3.85a
200	0.026a	23.6a	22.1a	0.91a	0.000a	3.66a
*Portulaca oleracea*						
0	19.0b	10.8b	10.9b	0.625c	0.000a	8.71b
50	21.4b	10.2b	10.0b	0.612c	0.124b	8.34b
100	15.1a	5.51a	5.75a	0.523b	0.132b	6.45ab
150	12.8a	4.88a	5.36a	0.487ab	0.092ab	5.71a
200	14.4a	4.08a	4.50a	0.425a	0.000a	7.11ab
*Salsola komarovii*						
0	14.7a	4.91a	4.36a	0.502c	n.d	3.89a
50	18.5c	3.91ab	3.82ab	0.493c	n.d	4.32a
100	19.4c	4.80ab	4.19ab	0.462b	n.d	5.51b
150	17.9bc	5.86b	4.66b	0.416a	n.d	5.96bc
200	15.8ab	5.86b	4.55b	0.408a	n.d	6.83c

n.d.—not detected. Different letters indicate statistically significant differences between means of salinity treatments according to one-way ANOVA Tukey’s test when *p* < 0.05.

**Table 2 metabolites-15-00724-t002:** The effect of hydroponic solution salinity (NaCl mM·L^−1^) on the accumulation of mineral elements in *Plantago coronopus*, *Portulaca oleracea* and *Salsola komarovii* leaves (mg·g^−1^ DW). Each cell color represents statistically significant differences between means of salinity treatments according to one-way ANOVA Tukey’s test when *p* < 0.05.

**NaCl**	**0**	**50**	**100**	**150**	**200**		**0**	**50**	**100**	**150**	**200**		**0**	**50**	**100**	**150**	**200**	**Macro elements**
**P**	0.7	0.7	0.8	0.8	0.8		0.52	0.76	1.3	0.8	1.31		0.4	0.53	0.6	0.7	0.8	
**K**	35	25	20	20	19		52.4	47.9	38	30	29.6		41.9	37.8	35	35	35	
**Ca**	22.3	13	11	11	8.2		5.67	3.79	3.4	2.4	2.67		15	8.49	5.7	4.7	5	
**Mg**	3.45	2.1	2.2	2.2	1.9		4.89	5.1	5.4	4.4	4.85		3.93	3.26	3.1	3.1	3.3	
**Fe**	162	167	194	185	169		56.1	47.6	36	33	24.3		35.5	51.3	29	33	26	**Micro elements**
**Mn**	115	99	112	120	101		208	274	237	188	196		136	111	106	100	107	
**Na**	5172	689	665	116	103		1659	1471	131	804	3875		1961	1112	679	670	551	
**Zn**	37.1	35	46	53	51		49.4	101	148	107	216		21.9	20.5	23	26	26	
** *Plantago coronopus* **						** *Portulaca oleracea* **							** *Salsola komarovii* **					
ANOVA between salinity					a		ab	b	bc	c	d	e						

## Data Availability

Data is available upon personal request.

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
