# Peer review of "Shift in Metabolite Profiling and Mineral Composition of Edible Halophytes Cultivated Hydroponically Under Increasing Salinity"

_metabolites, 2025, doi:10.3390/metabo15110724_

Round 1

Reviewer 1 Report

Comments and Suggestions for Authors

Manuscript is quite interesting and novel: exploring the potential halophytes response to the increased salt stress. Findings are relevant and significant for advancing the field. I have following suggestion for improvement: 

See botanical name should be in italic (figure, reference, other section of manuscript)

Keywords: Include the word nutrient; Arrange words in alphabetical order

Material and Method

After how many days of treatment exposure plant samples were analysed ?

Line 97: If these growing condtions (RH, Temp, Light) has been adopted from previous studies must cite them

Section 2.2: 

shift to section 2.1

Section must be devided into following seperate sections instead of one section: Weight related parameters (FW, DW) first
then  spectrophotometry based protein estimation 
Organic acids and Sugar estimation
Mineral nutrient estimation

In each section must describe the methodology clearly for scope of repeatability of such research by seeing used methods here.  Thats require for a science. 
Where ever requires cite the reference of published study from where manuscript has been taken

explicitly mention the name of minerals evaluated in study

Suggested to add PCA analysis for identifying key factors in the multi-species based study
It would give better picture how key composition shifts away/move closer to each other

Results and discussion

Suggest revising the correlation results for better clarity and detail

The correlation results are written in too general terms; by seeing the figure, hardly can one determine significant differences among all these parameters. 
Rather i suggest to revise the results: write separate for physiological, mineral nutrient and then correlation for physiological-mineral nutrient parameter. 
Provide results in seperate paragraph for each species
and then finally must see if any common trend  

The word significant has been used to refer a correlation significant and non-significant test, and the values with *, **, *** should be there (for different p values)
If not possible, here the correlation tables with exact values must be included in the supplementary file.

Study chosen edible halophyte species? however, in the discussion, I could hardly see the relevance of the composition obtained at different concentrations from a nutritional point of view 
Particularly in the micro nutrient Fe and Zn; Low Na diet 
I suggest to expand the discussion beyond plants philological response to different stress condition to the "change in nutritional values in response to stress condition" any recommendations if there should be a part of this, like cultivation of species beyond this salinity, leads to deterioration of nutritional values or a high Na diet... that would be more interesting and add value to the study 

Conclusions: if any limitation of present study and/or any future research suggestion based on this study must be included in a research to guide fellow researchers to expand beyond your findings.

Author Response

Response to Reviewer 1 Comments

Summary

Thank you very much for taking the time to review this manuscript. Please find the detailed responses below and the corresponding revisions and corrections in track changes in the re-submitted files. Graphical abstract was also added.

Comments 1: See botanical name should be in italic (figure, reference, other section of manuscript)

Response 1: Thank you for pointing this out. We agree with this comment. The botanical names are corrected in the figures in italic. This change can be found in figures 1, 2 and 4.

Comments 2: Keywords: Include the word nutrient; Arrange words in alphabetical order.

Response 2: Corrected, L:27-28

Comments 3: After how many days of treatment exposure plant samples were analysed?

Response 3: Thank you for pointing this out. L98-99: “Plant samples were analyzed after 3 weeks from germination for Plantago coronopus and Portulaca oleracea, and after 4 weeks for Salsola kamarovii.”

Comments 4: Line 97: If these growing conditions (RH, Temp, Light) has been adopted from previous studies must cite them

Response 4: The growing conditions were adopted from our previous works, citation added. L:106

Comments 5: Section 2.2: shift to section 2.1. Section must be devided into following seperate sections instead of one section: Weight related parameters (FW, DW) first

then spectrophotometry based protein estimation

Organic acids and Sugar estimation

Mineral nutrient estimation

Response 5: Thank you for pointing this out. Sections were divided as it was proposed. However, we decided to place the same order as it was. We believe that the essential information for the "Metabolites" journal focuses on metabolite changes, while information about biomass changes serves as complementary data.

Comments 6: In each section must describe the methodology clearly for scope of repeatability of such research by seeing used methods here.  Thats require for a science.

Where ever requires cite the reference of published study from where manuscript has been taken explicitly mention the name of minerals evaluated in study

Response 6: Agree. I have revised the methodology and improved it according to the suggestions. L:139-154

Comments 7: Suggested to add PCA analysis for identifying key factors in the multi-species based study

It would give better picture how key composition shifts away/move closer to each other

Response 7: PCA analysis was added. L:159-160; L:251-263

Comments 8: Suggest revising the correlation results for better clarity and detail

The correlation results are written in too general terms; by seeing the figure, hardly can one determine significant differences among all these parameters.

Rather i suggest to revise the results: write separate for physiological, mineral nutrient and then correlation for physiological-mineral nutrient parameter.

Provide results in seperate paragraph for each species

and then finally must see if any common trend 

Response 8: I revised the manuscript and improved it according to the suggestions.

Comments 9: The word significant has been used to refer a correlation significant and non-significant test, and the values with *, **, *** should be there (for different p values)

If not possible, here the correlation tables with exact values must be included in the supplementary file

Response 9: data were processed using one-way ANOVA, the Tukey‘s HSD and analyzed at the confidence level p < 0.05. There was no different p values presented.

Comments 9: Study chosen edible halophyte species? however, in the discussion, I could hardly see the relevance of the composition obtained at different concentrations from a nutritional point of view

Particularly in the micro nutrient Fe and Zn; Low Na diet

I suggest to expand the discussion beyond plants philological response to different stress condition to the "change in nutritional values in response to stress condition" any recommendations if there should be a part of this, like cultivation of species beyond this salinity, leads to deterioration of nutritional values or a high Na diet... that would be more interesting and add value to the study

Response 9: Thank you for pointing this out. It is known that, halophytes are salt-tolerant plants that thrive in habitats with highly saline (≥200 mM NaCl), some extreme halophytes survive even under 1200 mM NaCl (https://doi.org/10.1071/FP16025 ). Halophytes tolerate high levels of salt, but they are also able to complete their life cycle in a salt concentration of at least 200 mM NaCl and to achieve optimal growth levels at NaCl concentrations between 200 and 1000 mM (https://doi.org/10.1007/978-3-030-17854-3 ).

The aim of this research was to compare the potential of edible halophytes grown in saline nutrient solutions in hydroponic systems within salt-tolerant ranges. Thus, high salt concentrations or salinity stress were not evaluated or discussed.

Comments 9: Conclusions: if any limitation of present study and/or any future research suggestion based on this study must be included in a research to guide fellow researchers to expand beyond your findings.

Response 9: Corrected, please see L:375-382

4. Response to Comments on the Quality of English Language

Point 1: (x) The English is fine and does not require any improvement.

Reviewer 2 Report

Comments and Suggestions for Authors

Dear authors,

    How are you!

    Halophytes and salt-tolerant species have emerged as viable candidates for saline hydroponic cultivation,yet their agronomic performance and physiological responses lating hydroponic systems still require further investigation. The relationship between salinity levels, secondary metabolite biosynthesis, mineral element availability, and plant growth trade-offs under hydroponic conditions remains poorly defined too. Additionally, most nutrient solution formulations have been developed for glycophytic species, and leaving a gap in tailored nutrient regimes for salt-resilient crops, then leading to a lack of tailored mineral nutrition strategies for halophytes or salt-tolerant horticultural species. There is a critical need to develop crop- and context-specific models that account for physiological and metabolic responses under constrained water and nutrient scenarios.  All the questions mentioned above need researching, so this research is significant.

       However this paper needs major modification.

      1. The title is too extensive, suggest to limit the given 3 species.

      2. The keywords must be chosen from the title first, then suggest they can include the words such as Metabolite Profiling, Mineral Composition,Edible Halophytes,hydroponic systems, Salinity.

      3. Section abstract needs rewriting after modifying the full text.

    4. The last paragraph is unsuitable because this manuscript must answer to the questions mentioned after line 65 of section introduction.

     5. Under table 1, P≤ 0.05 is incorrect, and it must be p<0.05, and the order of English letters followed the means is incorrect too, and the order must be agreeable with the order of the means size.

      6. All figures are non-standard, and the axies are lost.

     7. Section results need adding the theoretical analysis, and some contents can be extracted from section discussion.

     8. Section discussion needs simplifying, and it can't replicate the contents of section results, and some theoretical analysis of the results can be transferred into section results, additionally the analysis of mechanism of salinity-resistance or tolerance needs adding, and not only comparing with the former reports.

      9. Section conclusion needs rewriting, and it must be sublimated up to the mechanism of salinity resistance or tolerance based on section discussion.

        10. Add sub-titles according to the contents in sections results and discussions.

Author Response

Response to Reviewer 2 Comments

Thank you very much for taking the time to review this manuscript. Please find the detailed responses below and the corresponding revisions and corrections in track changes in the re-submitted files. A graphical abstract was also added.

Comments 1: The title is too extensive, suggest to limit the given 3 species.

Response 1: Thank you for pointing this out. Considering the journal requirements for the title length, the authors decided to leave the current version. Adding the names of the 3 studied plants, each consisting of 2 words, would greatly expand it.

Comments 2: The keywords must be chosen from the title first, then suggest they can include the words such as Metabolite Profiling, Mineral Composition, Edible Halophytes, hydroponic systems, Salinity.

Response 2: Partially improved. Following the Instructions for Authors, the words in the title are automatically included in search results and were not repeated in keywords.

Comments 3: Section abstract needs rewriting after modifying the full text. The last paragraph is unsuitable because this manuscript must answer to the questions mentioned after line 65 of section introduction.

Response 3: Thank you for pointing this out. Abstract was revised to match the full text.

Comments 4: Under table 1, P≤ 0.05 is incorrect, and it must be p<0.05, and the order of English letters followed the means is incorrect too, and the order must be agreeable with the order of the means size.

Response 4: p≤0.05 was corrected to p<0.05. data were processed using one-way ANOVA, the Tukey‘s HSD test. Different letters indicate statistically significant differences between means of salinity treatments. The arrangement of data presented in the tables and figures followed the experimental design.

Comments 5: All figures are non-standard, and the axies are lost

Response 5: Thank you for pointing this out. All figures were corrected.

Comments 6: Section results need adding the theoretical analysis, and some contents can be extracted from section discussion.

Response 6: I revised the manuscript and improved it according to the suggestions. PCA analysis was added. L:159-160; L:251-263

Comments 7: Section discussion needs simplifying, and it can't replicate the contents of section results, and some theoretical analysis of the results can be transferred into section results, additionally the analysis of mechanism of salinity-resistance or tolerance needs adding, and not only comparing with the former reports.

Response 7: The discussion section was divided into two subsections explaining 4.1 metabolic response and 4.2 ion regulation and biomass allocation strategies in halophytes under increased salinity. During this research, we evaluated the contents of primary and secondary metabolites, mineral elements, and the shift in biomass accumulation rather than key players involved in the biochemical mechanisms. The need for an understanding of the mechanistic basis of species-specific osmotic adjustment in halophytes by integrating metabolomic and transcriptomic approaches is added in the last section. L:375-382

Comments 8: Section conclusion needs rewriting, and it must be sublimated up to the mechanism of salinity resistance or tolerance based on section discussion.

Response 8: Revised and corrected.

Comments 9: Add sub-titles according to the contents in sections results and discussions

Response 9: Revised and corrected.

Point 1: (x) The English is fine and does not require any improvement.

Reviewer 3 Report

Comments and Suggestions for Authors

Major concerns

  1. Results part should be divided into several parts with subtitles.
  2. Figure 1A should be divided with sugar and protein. Error bars should also be added into Figure 1.
  3. The part of results related to Figure 2 should be moved to the front in results part.
  4. The color in Table 2 should be changed with light colors to make numbers visible.
  5. The PCA analysis should be added to find key factors.
  6. The language should be edited by native speakers.

Author Response

Response to Reviewer 3 Comments

Thank you very much for taking the time to review this manuscript. Please find the detailed responses below and the corresponding revisions and corrections in track changes in the re-submitted files. A graphical abstract was also added.

Comments 1: Results part should be divided into several parts with subtitles.

Response 1: Thank you for pointing this out. Revised and divided.

Comments 2: Figure 1A should be divided with sugar and protein. Error bars should also be added into Figure 1.

Response 2: Figure 1 was divided into sugars and proteins, error bars added.

Comments 3: The part of results related to Figure 2 should be moved to the front in results part

Response 3: Thank you for pointing this out. Corrected.

Comments 4: The color in Table 2 should be changed with light colors to make numbers visible.

Response 4: Colors changed for better visibility.

Comments 5: The PCA analysis should be added to find key factors

Response 5: I revised the manuscript and improved it according to the suggestions. PCA analysis was added. L:159-160; L:251-263

Response 9: Revised and corrected.

Point 1: (x) The English is fine and does not require any improvement.

The language should be edited by native speakers

Point 1: The evaluation of English language differs, since the other two reviewers had no comments on the English language quality.

Round 2

Reviewer 1 Report

Comments and Suggestions for Authors

Manuscript has been improved as suggested. I have minor suggestion remaining as given below

Carefully see botanical name in references, must be italic (such as Ref No 19) and recheck carefully. 

As previously also suggested that: The correlation heatmap shown in Figure lacks the actual correlation coefficients and their significance levels. Without reporting r-values and corresponding p-values, it is not appropriate to conclude that the correlations are statistically significant. The authors are advised to include the numerical values of the correlation coefficients and indicate statistically significant correlations (e.g., p < 0.05) based on a t-test for Pearson correlation. Otherwise, remove the word "Significant" from the correlation results. 

Author Response

Thank you for your very valuable comments. Please find the detailed responses below and the corresponding revisions and corrections in track changes in the re-submitted files. Supplemental material was also added.

Comment: Carefully see botanical name in references, must be italic (such as Ref No 19) and recheck carefully. 

Response: The references were revised and corrected.

Comment: As previously also suggested that: The correlation heatmap shown in Figure lacks the actual correlation coefficients and their significance levels. Without reporting r-values and corresponding p-values, it is not appropriate to conclude that the correlations are statistically significant. The authors are advised to include the numerical values of the correlation coefficients and indicate statistically significant correlations (e.g., p < 0.05) based on a t-test for Pearson correlation. Otherwise, remove the word "Significant" from the correlation results. 

Response: The tables reporting r-values and corresponding p-values were added in the supplementary files.

Reviewer 2 Report

Comments and Suggestions for Authors

Dear authors,

      I hope you are doing well!

      According to your response, some comments last turn were ignored, please go on with the modifications based on the last comments 2~6.

Author Response

Thank you for your very valuable comments. Please find the detailed responses below and the corresponding revisions and corrections in track changes in the re-submitted files. Supplemental material was also added.

Comment: some comments last turn were ignored, please go on with the modifications based on the last comments 2~6.

Response:

  1. The keywords must be chosen from the title first, then suggest they can include the words such as Metabolite Profiling, Mineral Composition, Edible Halophytes, hydroponic systems, Salinity. - The keywords were added.
  2. Section abstract needs rewriting after modifying the full text. – The abstract was revised, L:12, L:23, L:25-31
  3. The last paragraph is unsuitable because this manuscript must answer to the questions mentioned after line 65 of section introduction. – The text was revised, L:86-89
  4. Under table 1, P≤ 0.05 is incorrect, and it must be p<0.05, and the order of English letters followed the means is incorrect too, and the order must be agreeable with the order of the means size. – P value corrected L:180. The order of English letters corresponds to the increase in size of the means in each column specifically for each observation. As indicated previously, the results are presented according to the increasing salinity treatment, because otherwise it is impossible to represent data in one table correctly. If the reviewer does not agree with my explanation, please provide a more detailed explanation for the expectation of data presentation.
  5. All figures are non-standard, and the axies are lost. – Non-standard figures were chosen to better represent the complexity of the experimental scheme. This does not object to the guidelines for authors. The lost axes titles were corrected in Figure 1C, and the information was added in Figure 2.

Reviewer 3 Report

Comments and Suggestions for Authors

The authors have addressed most of the comments. The manuscript could be accepted after grammar checking.

Author Response

Thank you for your valuable feedback. Additional supplemental material has also been included.